# The Effect of Fe/Al Ratio and Substrate Hardness on Microstructure and Deposition Behavior of Cold-Sprayed Fe/Al Coatings

**DOI:** 10.3390/ma16020878

**Published:** 2023-01-16

**Authors:** You Wang, Nan Deng, Zhenfeng Tong, Zhangjian Zhou

**Affiliations:** 1School of Materials Science and Engineering, University of Science and Technology Beijing, Beijing 100083, China; 2School of Nuclear Science and Engineering, North China Electric Power University, Beijing 102206, China

**Keywords:** cold spray, Fe/Al composite coatings, Fe/Al ratio, in situ hammer effect, substrate hardness

## Abstract

Fe/Al composite coatings with compositions of Fe-25 wt.% Al, Fe-50 wt.% Al and Fe-75 wt.% Al were deposited on pure Al and P91 steel plates by a cold spray, respectively. The microstructure of the cross-section of the fabricated coatings was characterized by SEM and EDX. The bonding strength between the coatings and substrates was measured and analyzed. The effects of the Fe/Al ratios and substrate hardness on the deposition behavior were investigated. It was interesting to find fragmented zones in all fabricated coatings, which were composed of large integrated Al particles and small fragmented Al particles. Meanwhile, the fraction of fragmented zones varied with the fraction of the actual Fe/Al ratio. An Fe/Al ratio of 50/50 appeared to be an optimized ratio for the higher bonding strength of coatings. The in situ hammer effect caused by larger and harder Fe particles played an important role in the cold spray process. The substrate with the higher hardness strengthened the in situ hammer effect and further improved the bonding strength.

## 1. Introduction

Fe/Al coatings are promising candidates for applications in extreme environments due to their excellent properties such as high-temperature strength, corrosion resistance, wear resistance and water vapor resistance [1,2,3,4,5,6]. Fe/Al coatings have been investigated for the surface protection of aerospace parts, as a tritium permeation barrier in fusion reactors [7], the surface protection of tubes in power plants and the surface protection of microelectronic elements [8]. Fe/Al coatings can be fabricated by different technologies, including spraying [9], hot dipping [10] and electron beaming [11]. Among them, a thermal spray is an efficient and economic method. However, the high operation temperature during thermal spraying may induce problems such as impurities (oxides and unknown iron aluminides), high porosities, thermal residual stress and even microcracks [12,13,14,15]. Brittle iron aluminides also easily form during high-temperature processes. These defects significantly deteriorate the properties of the thermal spray coatings.

In order to solve the problems caused by a high temperature during spraying, solid-state deposition technology—namely, a cold spray—has been developed in recent years, which can operate at temperatures much lower than the melting point of the sprayed materials [16]. It utilizes high-pressure compressed gas to propel microsized particles onto a substrate under atmospheric conditions. In this way, incident particles obtain high-velocity impact energy; the kinetic energy is then transferred into plastic deformation energy and thermal energy [17]. In most conditions, the effect of localized thermal softening is greater than working hardening, which leads to adiabatic shear instability [18]. Thus, by relying on the plastic deformation and adiabatic shear instability of both incident particles and substrates, powders can be deposited on and bonded with substrates by cold spraying. Cold spraying has been successfully applied to fabricating aerospace spars, stringers and frameworks as well as repairing damaged structural components [19,20,21,22].

In previous studies, powders with good plastic deformation abilities such as Al, Zn, Cu and Ni were commonly applied to cold spraying [23,24]. With the development of this technology, composite feedstock powders have been introduced into cold spraying in recent decades [13,25,26,27,28,29,30]. It was noticed that the composite feedstock powders were always composed of soft metals such as Al/Cu, Ti/Al, Ni/Cu and Al/Mg, among others [26,30,31]. Thus, it is interesting to investigate the cold spray deposition behavior of feedstock powders mixed with both soft and hard metals, which may show better mechanical properties and a more promising application prospect.

Compared with soft metal powders (such as Al), Fe powders are difficult to deposit by cold spraying because of the high elastic modulus. The addition of Al powders into the feedstock powders may improve the deposition efficiency of Fe powders. Previous studies about Fe/Al cold-sprayed coatings mostly used an Fe/Al alloy as a feedstock powder [32]. However, dual-phase Fe/Al composite powders are rarely applied in cold spraying as feedstock powders. Wang et al. [9,33] used Fe/Al composite powders as feedstock powders to investigate the coating characteristics and phase transformations during heat treatments. However, the effect of different compositions of Fe/Al coatings have not been widely discussed. Substrates with different hardnesses may influence the effect of cold spraying [17,18]. In addition, Fe/Al coatings are usually sprayed by helium and nitrogen to improve the deposition efficiency [9,33,34], but they are expensive. It is interesting to understand whether Fe/Al coatings can be cold-sprayed using the atmosphere as a propellent gas.

In this work, the atmosphere was used as a cold-sprayed propellent gas in the process of depositing Fe/Al composite coatings onto pure Al and P91 steel substrates, respectively. The effects of the Fe/Al ratio and substrate hardness on the deposition behavior and microstructure of the coatings were investigated. The bonding strength was measured and the bonding mechanism was discussed.

## 2. Experimental Procedure

### 2.1. Cold Spraying

Commercial Fe (99.99 wt.% purity, -200 mesh) and Al (99.99 wt.% purity, -325 mesh) powders were used as the feedstock powders. Most of the Fe and Al powders were spherical, as shown in Figure 1.

Different ratios of Al powders—25 wt.%, 50 wt.% and 75 wt.%—were mechanically mixed with Fe powders in a V-type mixer for 15 h. The mixed Fe/Al powders were preheated by a propelling gas at 350 °C and deposited by a TECHNY LP-TCY-III supersonic cold spraying system onto the substrates. The converging–diverging nozzle of the gun had a throat diameter of 6 mm. The atmosphere gas worked as the propelling and powder-feeding gas. Commercial P91 steel and pure Al plates were used as the substrates. The composition of the commercial P91 steel was Fe-9.00Cr-1.00Mo-0.30Mn-0.20V-0.08C (wt.%) with a hardness of 180 HV; the hardness of the commercial pure Al plates was 85 HV. The constituents of all samples used in this study are listed in Table 1.

### 2.2. Shear Bonding Strength Test

The bonding strength was measured by a specially designed shear bonding test, as shown in Figure 2. First, a bar with a half-coating and a half-substrate was cut from a cold-sprayed sample with a diameter of 6 mm and a length of 15 mm, as shown in Figure 2a. Careful wire cutting was then performed from the surface of coating to the coating/substrate interface to form a gap, as well as from the surface of the substrate to the coating/substrate interface to form another gap, as shown in Figure 2b. The distance between these two gaps was 1 mm, as marked by a red line in Figure 2c, which is the front view of Figure 2b. In this way, the whole bar could be considered to be composed of two parts, A and B, connected by the red line marked in Figure 2c. Finally, A and B were separated by applying two forces at both ends of the bar by a CSS-WAW electro-hydraulic servo universal testing machine. The force F was used to calculate the shear bonding strength according to Equation (1):P = F/S(1)

P is the shear bonding strength, MPa; F is the force, N; and S is the contact area, m^2^. In this test, S = 1 × 10^−3^ × 6 × 10^−3^ = 6 × 10^−6^ m^2^. The shear bonding strength tests were repeated three times for each specimen to ensure the accuracy of results.

### 2.3. Characterization

The as-sprayed coated samples were metallographically polished. The microstructure was investigated using field-emission scanning electron microscopy (SEM) (Zeiss Gemini 300 Ultra, Berlin, Germany) equipped with an X-ray spectrometer (EDS) (Oxford Xplore 30 energy-dispersive, Oxford, UK). The fracture surface morphology after the shear bonding strength test was investigated by SEM. An EDS elemental mapping analysis and a line scan analysis were used to measure the compositions. The actual area fractions of the Fe and Al particles in the coatings were estimated by ImageJ Software (Version 1.53, National Institutes of Health, Bethesda, MD, USA) using more than 10 BSE images, which covered the whole cross-section of the specimens. The average area fractions were roughly equal to the volume fractions (vt.%) of the Fe and Al particles in the coatings. They were finally transformed into weight fractions (wt.%) according to m = ρV.

## 3. Results and Discussion

### 3.1. Microstructure of the Fe/Al Coating

Figure 3 and Figure 4 show the cross-section morphologies of the Fe/Al coatings with different compositions. In all images, the white particles represent Fe and the grey particles represent Al. These two types of particles were homogenously distributed and showed obvious deformations, as seen in Figure 3a–c. It was also interesting to find that a few dark areas existed, typically along the inter-particle interface, with a different morphology from the porosities or inter-particle boundaries, as indicated by the red box in Figure 3a–c. Two different morphologies, flat zones and rough zones, that contained a number of small fragmented particles less than 100 nm could be observed in these dark areas at higher magnifications, as shown in Figure 3d–f and Figure 4a. The fragmented particles were identified as Al particles from the EDS spot analysis, as shown in Figure 3d–f. Furthermore, the EDS elemental mapping analysis on Fe-75Al-Al confirmed that only the Al element appeared in the whole of the dark areas, which is clearly shown in Figure 4b. The EDS line scan analysis shown in Figure 4c further demonstrated that the content of the Al element in the left flat zone was lower than that in the right rough zone. It should be noted that the results of the line scan analysis were qualitative rather than quantitative due to a low count rate as the black area was lower than the surrounding area. This suggested that the dark areas were composed of Al particles and that they performed with two different morphologies, large integrated particles, and small fractured particles. 

This type of morphology (named the fragmented zones by the authors) could have been caused by the high strain rate of the Al particles and the in situ hammer effect of the Fe particles. It has been clarified in previous studies that smaller particles are easier to accelerate and obtain a much higher strain rate [28,31,33]. The Al powders used in this work had a smaller particle size; thus, the high strain rate easily broke the outer layer into fractions. The in situ hammer effect of the Fe particles also promoted this process, which has been confirmed in previous studies [35,36,37,38]. Moreover, a high strain rate can aggravate the effect of working hardening. When the effect of working hardening is higher than thermal softening, adiabatic shear instability is weakened, which leads to a loose bonding of these broken Al particles. Therefore, Al particles with fractured outer layers in fragmented zones are easy to peel off and retain a flat and concave morphology, as shown in Figure 4a.

A typical fracture morphology is shown in Figure 5a, in which one large particle with a crater at the upper right was adhered to several small particles. According to the EDS elemental mapping analysis in Figure 5b–d, it was deduced that this was a large Fe particle attached to a few small Al particles. Regarding the crater, there was no direct evidence to identify it as either an Al particle or an Fe particle that had peeled off from the large particle and left a crater. This type of fracture morphology could be attributed to loosen bonding caused by a high strain rate, as mentioned above.

### 3.2. Effect of the Addition of Al on the Deposition Behavior of the Fe/Al Coating

Pure Fe powders are very difficult to be cold-sprayed onto an Al substrate, especially when using atmosphere as the propellent gas, as reported by previous work [34]. Al powders are one of the most common and efficient materials used in cold spraying due to their high ductility and low elastic modulus [25,30]. The addition of Al powders enabled the dual-phase metal matrix composite powders to form the Fe/Al coating deposited on the Al substrate. The actual fractions of Fe, Al and the fragmented zones were calculated by ImageJ, as mentioned in Section 2.3. Among these, the fraction of the fragmented zones—including large integrated Al particles and small fractured Al particles—was used to indicate the quality of the inter-particle bonding of the cold-sprayed Fe/Al coatings. According to the data listed in Table 2, the actual Fe/Al ratios of Fe-25Al-Al and Fe-50Al-Al were close to the designed compositions whereas the ratio of Fe-75Al-Al was higher than the designed composition, indicating a greater loss of Al compared with the other two during cold spraying. Meanwhile, the fraction of fragmented zones decreased from 10.22% to 7.75%, then rose to 9.62% along with the increase in Al. This trend could be attributed to the in situ hammer effect of the Fe powders [35,36,39]. The density of Fe is much larger than that of Al. The harder and larger Fe powders in the composite feedstock powders hammered the Al powders into small and fractured particles, as mentioned above; thus, the fraction of fragmented zones decreased with the lower actual fraction of Fe powders in Fe-25Al-Al and Fe-50Al-Al. 

It was noted that the fraction of Fe was much higher than the designed composition in Fe-75Al-Al and the fraction of fragmented zones increased again when the content of Al increased from 50% to 75%. This result was quite different from expectation, as Fe powders ought to bounce off during cold spraying because of the higher hardness and larger size [40]. However, it might have been a result of the addition of 75 wt.% Al. The Fe powders may have been harder to bounce off when surrounded by sufficient soft Al powders compared with the other two samples. This was in agreement with the experiment of Ng et al. [31]. They found that following a rise in the Al content of samples, Ti6Al4V was tightly surrounded by Al; thus, the ratio of rebound Ti alloy particles decreased. It seemed that the addition of Al effectively prevented the rebound behavior of the Fe powders. Thus, the Fe/Al ratio of Fe-75Al-Al and the fraction of fragmented zones were higher than those of Fe-50Al-Al.

As a result, the effect of the addition of Al powders worked in two ways: the in situ hammer effect and the rebound behavior of the Fe powders. The actual fraction of Fe and fragmented zones had the same trend. In order to ensure a successful deposition and to reduce the loss of feedstock powders, an optimized ratio of Fe/Al during cold spraying on a pure Al substrate is around 50/50.

### 3.3. Effect of the Substrate Hardness on the Microstructure

Figure 6 shows macro pictures of the as-sprayed samples with different substrates after wire cutting and polishing, in which the coating thickness of Fe-75Al-Al was obviously much lower than the other two samples. This was probably related to its special deposition behavior, as discussed before. It should be noted that the top surface of these coatings was non-uniform because our priority was to spray the coatings as thickly as possible during fabrication rather than obtain a flat surface. In general, it is possible to obtain a rather high thickness at the millimeter level for cold-sprayed Fe/Al coatings. Furthermore, the coatings with a pure Al substrate integrated well whereas the coatings of Fe-50Al-P91 and Fe-75Al-P91 peeled off from the P91 steel substrate during wire cutting. It seemed that only the Fe-rich coating could be successfully deposited onto the P91 steel substrate, indicating that the substrate hardness might have an effect on the deposition behavior.

The Fe-25Al-Al and Fe-25Al-P91 samples were compared to investigate the effect of the substrate hardness on the deposition behavior, including the microstructure and bonding strength. Figure 7, Figure 8, Figure 9 and Figure 10 show the SEM images of the Fe-25Al-Al and Fe-25Al-P91 samples. The Fe/Al coatings were dense and the Fe and Al particles were homogeneously distributed, as shown in Figure 7. The fraction of fragmented zones in Fe-25Al-P91 appeared to be much smaller than that in Fe-25Al-Al. The fraction of Al fragmented zones in Fe-25Al-P91 was also calculated by ImageJ Software, as mentioned in Section 2.3, and is listed in Table 2. The value was only 1.16%, which was much lower than the 10.22% of Fe-25Al-Al. It was also reflected by the morphology in Figure 7, in which less fragmented Al particle zones can be seen in Figure 7b than in Figure 7a. The fracture morphology in Figure 8 shows this feature more directly; the light particles are Fe and the dark particles are Al. It was apparent that the Fe particles in Fe-25Al-P91 deformed much more than the other sample. In previous studies, many researchers believed that the hardness of the substrate could only constrain a layer that was several micrometers thick [16,17,30]. Wang [41] also proved that a harder substrate was of benefit when strengthening the inter-particle bonding in an Al_2_O_3_ coating. The decrease in fragmented zones in Fe-25Al-P91 and a greater number of deformed Fe particles appeared to be further evidence to support this conclusion. Fragmented zones appeared along not only the inter-particle interface but also the coating/substrate interface of Fe-25Al-Al and a narrow gap existed along the coating/substrate interface of Fe-25Al-P91, as shown in Figure 7. Further analyses of the coating/substrate interfaces of the Fe-25Al-Al and Fe-25Al-P91 samples are shown in Figure 9 and Figure 10, respectively.

It was noticed that most of the small and fractured Al particles had peeled off and left integrated particles in the fragmented zones along the coating/substrate interface, as shown in Figure 9a. A few microcracks originated and expanded only in these zones, as indicated by the red arrows. The SEM images and EDS elemental mapping analysis results shown in Figure 9b–d proved that the composition of the fragmented zones along the interface was also Al elements. It appeared that these fragmented zones along the interface in Fe-25Al-Al were not continuous. In contrast, the narrow gap, with a less than 1 μm width, in Fe-25Al-P91 was continuous along the interface, as shown in Figure 10b. The EDS line scan analysis showed both small fractured Fe and Al particles existed in the gap, as shown in Figure 10a. We believed that the formation of these two different morphologies along the interface was induced by the in situ hammer effect. The authors believed that the in situ hammer effect could be divided into two parts: hammering and densifying. Regarding Fe-25Al-Al, the hammering effect was strong due to the high content of Fe powders, but the densifying effect was relative weak because of the soft Al substrate. Thus, microcracks originated in the fragmented zones along the coating/substrate interface, which might have been caused by the accumulated stress. Regarding Fe-25Al-P91, the cooperated effect of hammering and densifying promoted the formation of the gap consisting of Fe and Al particles. During the process of cold spraying, high-speed Al powders reached the interface first. The Fe powders then crushed the Al particles into nanosized fragments by hammering. Finally, cooperating with the hard P91 steel substrate, the subsequent Fe powders densified the fragmented Al particles into a narrow gap at the interface, which worked as a transition layer to further improve the bonding strength. The densifying effect was weakened in the samples with fewer Fe particles (such as Fe-50Al-P91 and Fe-75Al-P91) with peeling-off coatings. In addition, due to the high hardness of both the Fe powders and P91 substrates, the high-speed Fe powders were easier to bounce off whilst being sprayed onto the P91 substrate than that onto the Al substrate. The effect of in situ hammering was weakened and led to the peeling off of the coatings on Fe-50Al-P91 and Fe-75Al-P91, as shown in Figure 6. In this way, it could be deduced that the substrate hardness had an impact on the deposition behavior.

### 3.4. Effect of the Fe/Al Ratio and Substrate Hardness on the Bonding Strength

The results of the bonding strength tests are listed in Table 3, in which the value of the bonding strength was the average value of three repeated tests. The variation trend of the bonding strength was the same as the trend of the actual Al fraction in Fe-25Al-Al, Fe-50Al-Al and Fe-75Al-Al. It may have been a result of the in situ hammer effect on the coating/substrate interface, including hammering and densifying, as mentioned in Section 3.3. The addition of Al significantly decreased the hammering effect and thus decreased defects such as microcracks along the coating/substrate interface, as shown in Figure 9a. Therefore, the bonding strength was improved. Regarding the Fe-25Al-P91 sample, its bonding strength of 109 MPa was higher than that of Fe-25Al-Al. However, with an increasing content of Al, the densifying effect was weakened and the bonding strengths of the coatings of Fe-50Al-Al and Fe-75Al-Al decreased. The fracture morphology along the coating/substrate interface after the shear bonding strength test is shown in Figure 11. Traces of the dropped Fe particles on the interface can be observed, marked by white dashed lines in Figure 11a, whereas the fragmented Al particles remained after the shearing testing, as shown in Figure 11b. This type of fracture morphology might be attributed to the higher hardness of the P91 steel substrate as the higher hardness of the substrate strengthened the densifying effect induced by the in situ hammer effect. During the process, the narrow gap at the interface of Fe-25Al-P91, as shown in Figure 10b, acted as the transition layer to further improve the bonding strength. 

## 4. Conclusions

Fe/Al coatings with different composition designs were fabricated onto pure Al and P91 steel substrates by cold spraying. The effects of the Fe/Al ratio and substrate hardness on the deposition behavior and bonding strength were investigated. The main conclusions can be summarized as follows:Samples with three different Fe/Al ratios all showed a special morphology of fragmented zones with Al elements. The fragmented zones were composed of large integrated Al particles and small fractured Al particles.The Fe/Al ratio showed a significant influence on the deposition behavior of the cold-sprayed coatings. Fe/Al coatings with different Fe/Al ratios could be successfully deposited onto an Al substrate by cold spraying; a coating with an Fe/Al ratio of 50/50 (wt.%) showed a relatively high bonding strength. For the P91 substrate, only a coating with an Fe/Al ratio of 75/25 (wt.%) could be successfully deposited.The hardness of the substrate had an obvious influence on both the deposition behavior and the bonding strength. Influenced by the in situ hammer effect, the fragmented zones were densified to a transition layer and further improved the bonding strength of Fe-25Al-P91.

## Figures and Tables

**Figure 1 materials-16-00878-f001:**
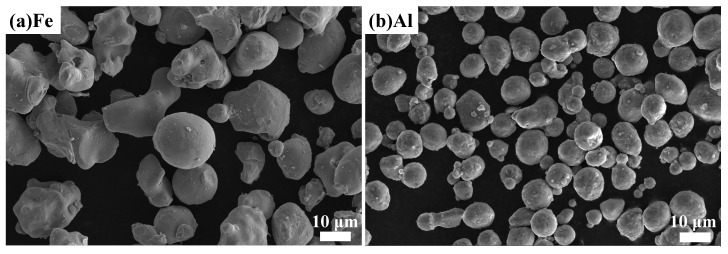
SEM images of feedstock powders: (**a**) Fe and (**b**) Al powders.

**Figure 2 materials-16-00878-f002:**
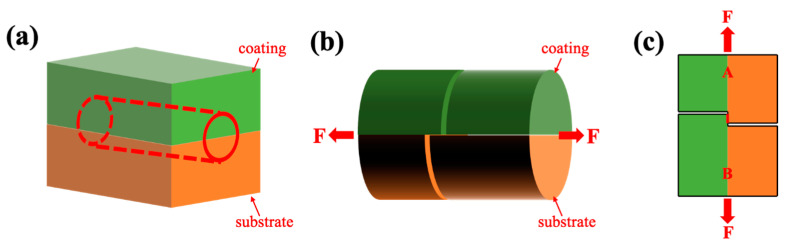
Scheme of shear bonding test, in which the green area represents the Fe/Al coating and the orange area represents the substrate; the bar in (**b**) was cut from (**a**) and the rectangle in (**c**) is the front view of (**b**).

**Figure 3 materials-16-00878-f003:**
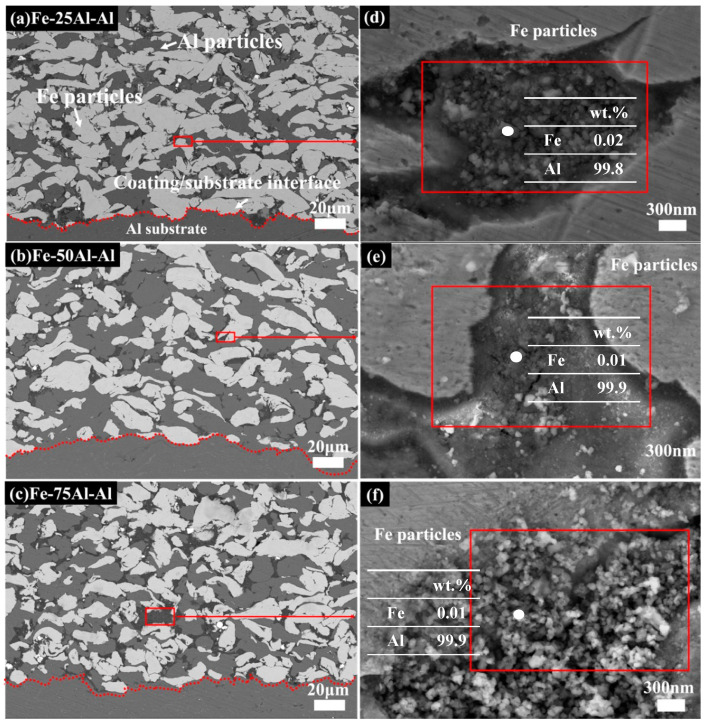
BSE images of cross-section of samples (**a**–**c**) and partial enlarged SEM images (**d**–**f**) with the results of the EDX spot analysis.

**Figure 4 materials-16-00878-f004:**
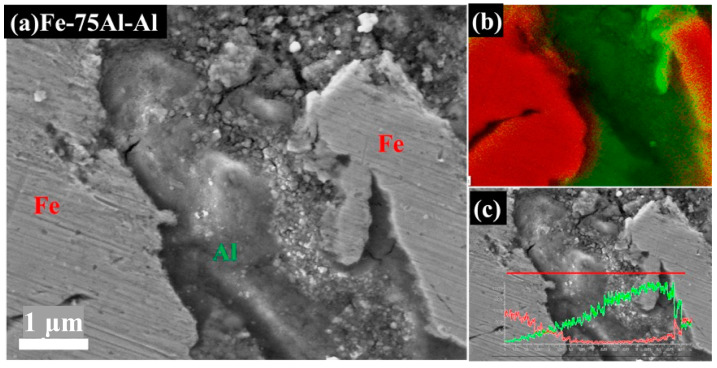
(**a**) SEM image of Fe-75Al-Al; (**b**) EDS mapping elemental analysis (red region is Fe and green region is Al); (**c**) line scan analysis (red and green lines represent the variations in the Fe content and Al content, respectively).

**Figure 5 materials-16-00878-f005:**
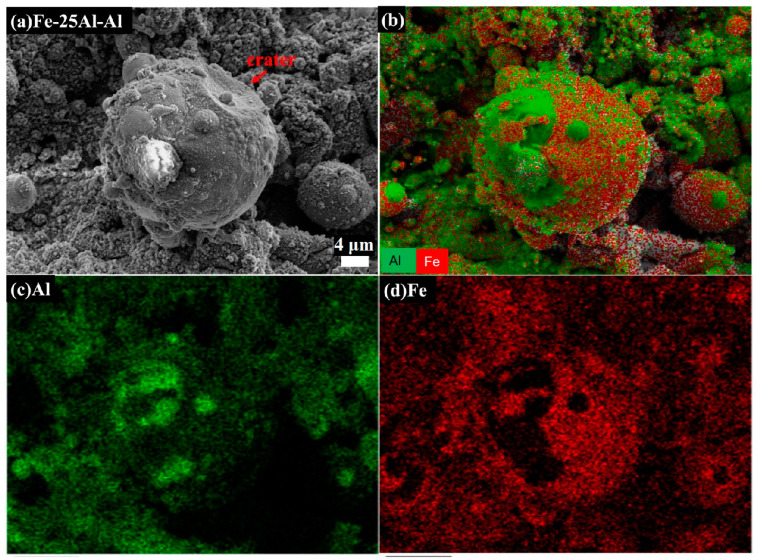
SEM image (**a**) and EDS elemental mapping analysis (**b**–**d**) of an Fe particle attached to many Al particles in the fracture morphology.

**Figure 6 materials-16-00878-f006:**
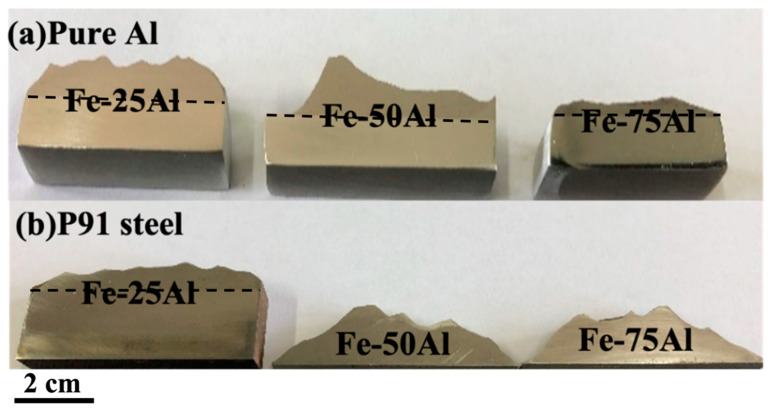
Images of as-sprayed samples after wire cutting and polishing: (**a**) samples with pure Al substrate; (**b**) samples with P91 steel substrate.

**Figure 7 materials-16-00878-f007:**
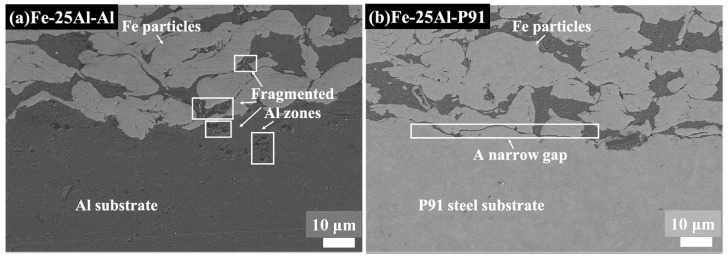
SEM images of Fe-25Al-Al and Fe-25Al-P91, in which fragmented zones appeared along the inter-particle interface and the coating/substrate interface in (**a**). A narrow gap existed along the coating/substrate interface in (**b**).

**Figure 8 materials-16-00878-f008:**
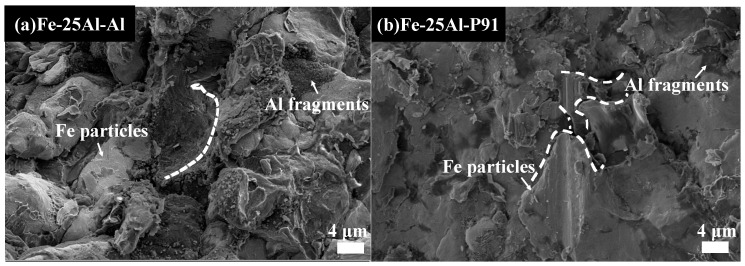
SEM images of the fracture morphology along the inter-particle interface of Fe-25Al-Al and Fe-25Al-P91.

**Figure 9 materials-16-00878-f009:**
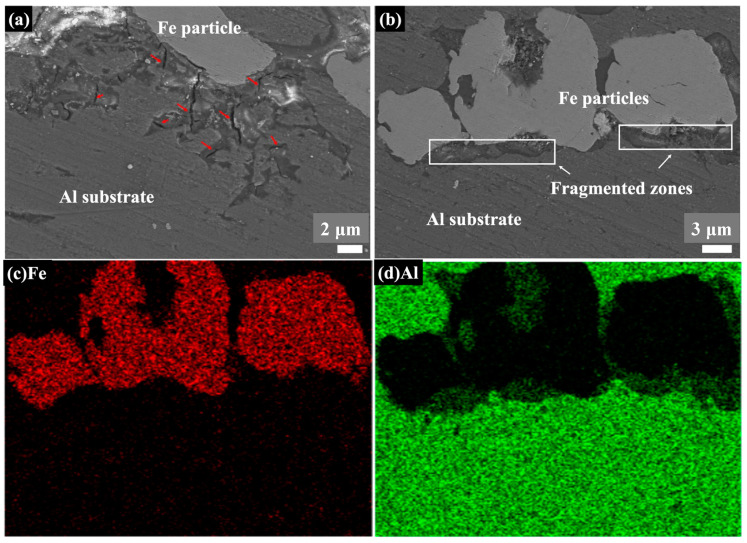
SEM images and EDS elemental mapping along the coating/substrate interface of Fe-25Al-Al; microcracks originated and expanded in the fragmented zones in (**a**) whereas non-continuous fragmented zones formed along the interface in (**b**–**d**).

**Figure 10 materials-16-00878-f010:**
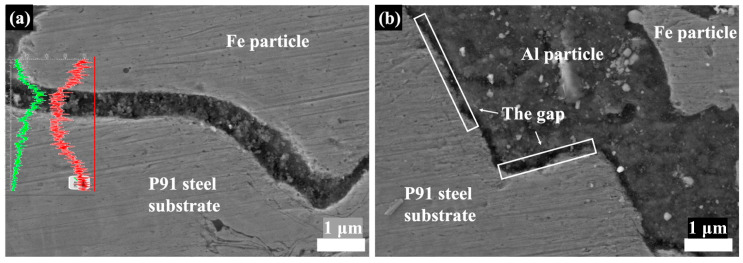
SEM images and EDS line scan analysis of coating/substrate interface of Fe-25Al-P91. (**a**) The green line is the trend curve of Al content and the red line is the trend curve of Fe content. (**b**) shows the enlarged images of the gap.

**Figure 11 materials-16-00878-f011:**
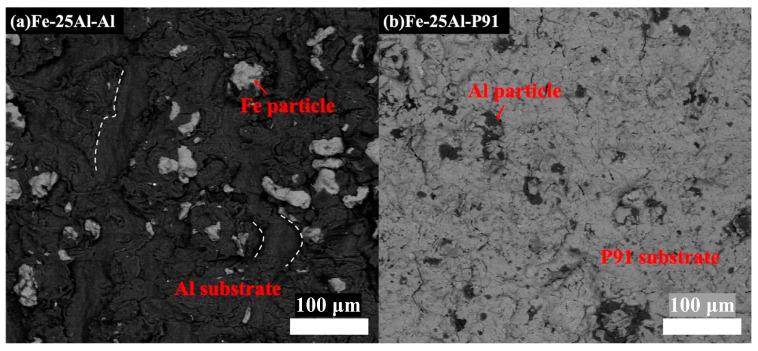
SEM images of fracture morphology along the coating/substrate interface of Fe-25Al-Al and Fe-25Al-P91 after shear bonding tests.

**Table 1 materials-16-00878-t001:** Constituents of samples (wt.%).

Samples	Fraction of Fe	Fraction of Al	Substrate
Fe-25Al-Al	75	25	Al
Fe-50Al-Al	50	50	
Fe-75Al-Al	25	75	
Fe-25Al-P91	75	25	P91
Fe-50Al-P91	50	50	
Fe-75Al-P91	25	75	

**Table 2 materials-16-00878-t002:** Actual fraction of Fe, Al and fragmented zones in Fe/Al coatings (wt.%).

Samples	Actual Fraction			
	Fe	Al	Fragmented Zones	Error
Fe-25Al-Al	60.45	29.33	10.22	±2.01
Fe-50Al-Al	43.28	48.97	7.75	±1.43
Fe-75Al-Al	45.56	44.82	9.62	±1.77
Fe-25Al-P91	66.89	31.95	1.16	±1.25

**Table 3 materials-16-00878-t003:** A summary of the actual fraction and bonding strength of all samples.

Samples	Actual Fraction (wt.%)			Bonding Strength (MPa)
	Fe	Al	Error	
Fe-25Al-Al	60.45	39.55	±2.01	52
Fe-50Al-Al	43.28	56.72	±1.43	73
Fe-75Al-Al	45.56	54.44	±1.77	65
Fe-25Al-P91	66.89	33.11	±2.13	109

## Data Availability

The data that support the findings of this study are available on request from the corresponding author, Z. Z., upon reasonable request.

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
