# Peer review of "The Effect of Fe/Al Ratio and Substrate Hardness on Microstructure and Deposition Behavior of Cold-Sprayed Fe/Al Coatings"

_materials, 2023, doi:10.3390/ma16020878_

Round 1
Reviewer 1 Report
In the present manuscript, the influence of Fe/Al ratio of feedstock powder on the deposition behavior, the microstructure and the bonding strength of the coating. However, I have some important concerns that should be addressed before the manuscript may be accepted:
General comments:
1) I found some inappropriate references. For example, on p.1 line 29, the authors described “Fe/Al coatings can be fabricated by different technologies, including spraying [9], electroplate [10], magnetron sputtering [11].” However, Ref.[9] is not an article on the cold sprayed Fe/Al coating, but on the polymer coating. Ref. [10] could not be found in the 2010 International Conference on Measuring Technology and Mechatronics Automation. Ref. [11] is not on the magnetron sputtering, but on the hot-dipping. Appropriate references must be cited throughout the manuscript.
2) In Section 2.1, Table 1 must be shown. I could not find it in the manuscript.
3) In Fig.4 (c), the line scan analysis indicated both the contents of Al and Fe elements are low in the left flat zone. Does this mean that other elements exist in the flat zone? Please explain about it in Section 3.1.
4) On p.7 lines 195-197, the authors claimed that “the optimized ratio of Fe/Al for an enhanced cold-sprayed Fe/Al coating on pure Al substrate should be around 50/50.” However, the properties of the coating such as strength, corrosion resistance, and wear resistance, depend on the Fe/Al ratio. Therefore, I think that the optimized ratio cannot be determined only by the result of the actual fraction.
5) The result of the hardness test explained in Section 2.3 must be shown in Section 3. I could not find it in the manuscript.
6) On p.11 lines 282-284, the authors said “The Al particles embed into the P91 substrate more tightly than Fe particles embed into the Al substrate ash shown in Fig. 12.” However, I could not confirm that thing from Fig.12. The authors should explain about it in more detail, or show more appropriate figures.
7) In Fig.7, the coatings of Fe-50Al-P91 and Fe-75Al-P91 were peeled off from the substrate. On the other hand, Fe-25Al-P91 indicated the highest bonding strength. Why were the coatings of Fe-50Al-P91 and Fe-75Al-P91 peeled off? Please discuss about it.
8) If it is possible, please show the result of the deposition efficiencies of each coating.
Detailed comments:
9) In Section 2.1, the size of the substrates and the hardness should be indicated. The authors discussed the influence of the substrate hardness on the deposition behavior. The information of the substrate hardness is important.
10) In Section 2.2, the size of the specimen used for the shear bonding strength test should be indicated. In Fig. 7, the coating thickness of Fe-75Al-Al seems to be not enough for the shear bonding strength test.
11) In Fig. 7, a scale must be shown.
Reviewer 2 Report
This manuscript the Fe/Al composite coatings with three different compositions of Fe-25 wt.% Al, Fe-50 wt.% Al and Fe-75 9 wt.% Al deposited on pure Al and P91 steel plates by cold spraying. The composition, microstructure, and bonding strength of the coatings were detailed studied. The manuscript is well organized. There are still some issues that need to be further addressed.
1. Different ratios of Al powders, 25 wt.%, 50 wt.% and 75 wt.%, were mechanical mixed 80 with Fe powders in a V-type mixer for 15 hrs. What does the author mean by mechanical mixing? Is there already some reaction of the powers or change of the shape?
2. 2.2. Shear bonding strength test. Please give the size of the bar and how exactly the bonding strength was calculated? Is this a standard method, otherwise please explain?
3. The hardness of coatings was measured using a 430SVD Vickers. However, in the manuscript there are not reported values for the coatings. Indeed, it is interesting to compare the hardness of the Al, Fe and the fragmented zones.
4. In Fig.3, the composition of the fragmented zones are not very clear. Please give more results, like EDS point analysis or small area mapping results to confirm.
5. The Table 2 and 3 are confusing. In Table 2, the composition was given by wt.%, but in Table 3 it was just %. How did the author obtain the values with wt.%? I assume that the image analysis can only give you area percent. Please clarify. Here also it is better to give the uncertainties of each composition.
6. Fig. 6 is actually presenting the same results as in Table 2, and can be deleted.
7. It should be noted that these samples were all with non- uniform coatings because the authors tried to find out the upper limit of coating thickness with different compositions and substrates. This sentence is unclear and very confusing. The coating looks non-uniform even in one single substrate. How could this happen and then the author needs to give more details in the experimental part. How was the exact deposition parameters for each substrate and how this affect the coating thickness?
8. Fig.7, please mark the interface between substrate and coatings.
9. Meanwhile the particles in Fig. 8 (b) deform better than that in Fig. 8 (a). The reviewer could not see how the particles in Fig. 8 (b) deform better than that in Fig. 8 (a)? how the author define the deform level? Then the stain in the coatings/particles should be measured.
10. The explanation on the different bonding strengths is not convincing. How the hardness affects the coating density and bonding was not clearly explained. In Fig.12 it is hard to see how the embedding of the particles into the substrates. Please give better images and revise this part.
Reviewer 3 Report
- Please ensures that your manuscript meets materials style requirements.
- Sort the keywords according to alphabetical order.
- The Abstract part needs to be improved. And the authors need to describe the main results in the abstract.
- Author mention in table 1 in line 8 but I did found the table
- To demonstrate the research gaps the current study aims to address, previous studies linked to it need to be explained in the introduction, including their work, novelty, and limitations.
- The conclusion part part needs to be improved. And the authors need to describe the main results in the abstract
Round 2
Reviewer 1 Report
This paper can be accepted after minor revisions:
1) Fig. 8 (b) shows the image of Fe-75Al-P91. However, Fe-25Al-Al and Fe-25Al-P91 should have been compared in this section. The image of Fe-25Al-P91 should be shown in Fig. 8.
2) Several typos can be found, especially in the abstract.
Author Response
Dear editors and reviewer,
We would like to thank you for your valuable comments concerning our article. Point-by-point responses to the reviewer are listed below.This paper can be accepted after minor revisions:
1) Fig. 8 (b) shows the image of Fe-75Al-P91. However, Fe-25Al-Al and Fe-25Al-P91 should have been compared in this section. The image of Fe-25Al-P91 should be shown in Fig. 8.
RESPONSE: We appreciate reviewer's careful reading. We should apologize for the wrong captions of Fig. 8. After carefully checking, we found Fig. 8 (b) showed the image of Fe-25Al-P91 rather than Fe-75Al-P91. All the captions in this revision manuscript were checked. Thanks for reviewer's kindly reminder.
2) Several typos can be found, especially in the abstract.
RESPONSE: We appreciate reviewer's kindly reminder. We have checked the manuscript and revised several typos. We hope the typos have been corrected at this time.
Above all, we really appreciate the reviewer’s constructive comments on this manuscript. We really hope the typos and other mistakes are corrected at this time.